# Counterfactual Invariance to Spurious Correlations: Why and How to Pass Stress Tests

Victor Veitch[1,2], Alexander D'Amour[1], Steve Yadlowsky[1], and Jacob Eisenstein[1]

[1]*Google Research*
[2]*University of Chicago*

## Abstract

Informally, a 'spurious correlation' is the dependence of a model on some aspect of the input data that an analyst thinks shouldn't matter. In machine learning, these have a know-it-when-you-see-it character; e.g., changing the gender of a sentence's subject changes a sentiment predictor's output. To check for spurious correlations, we can 'stress test' models by perturbing irrelevant parts of input data and seeing if model predictions change. In this paper, we study stress testing using the tools of causal inference. We introduce *counterfactual invariance* as a formalization of the requirement that changing irrelevant parts of the input shouldn't change model predictions. We connect counterfactual invariance to out-of-domain model performance, and provide practical schemes for learning (approximately) counterfactual invariant predictors (without access to counterfactual examples). It turns out that both the means and implications of counterfactual invariance depend fundamentally on the true underlying causal structure of the data—in particular, whether the label causes the features or the features cause the label. Distinct causal structures require distinct regularization schemes to induce counterfactual invariance. Similarly, counterfactual invariance implies different domain shift guarantees depending on the underlying causal structure. This theory is supported by empirical results on text classification.

## 1 Introduction

Our focus in this paper is the sort of spurious correlations revealed by "poke it and see what happens" testing procedures for machine-learning models. For example, we might test a sentiment analysis tool by changing one proper noun for another ("tasty Mexican food" to "tasty Indian food"), with the expectation that the predicted sentiment should not change. This kind of perturbative stress testing is increasingly popular: it is straightforward to understand and offers a natural way to test the behavior of models against the expectations of practitioners [Rib+20; Wu+19; Nai+18].

Intuitively, models that pass such stress tests are preferable to those that do not. However, fundamental questions about the use and meaning of perturbative stress tests remain open. For instance, what is the connection between passing stress tests and model performance on prediction? Eliminating predictor dependence on a spurious correlation should help with domain shifts that affect the spurious correlation—but how do we make this precise? And, how should we develop models that pass stress tests when our ability to generate perturbed examples is limited? For example, automatically perturbing the sentiment of a document in a general fashion is difficult.

The ad hoc nature of stress testing makes it difficult to give general answers to such questions. In this paper, we will use the tools of causal inference to formalize what it means for models to pass stress tests, and use this formalization to answer the questions above. We will formalize passing stress tests as *counterfactual invariance*, a condition on how a predictor should behave when given

35th Conference on Neural Information Processing Systems (NeurIPS 2021).

certain (unobserved) counterfactual input data. We will then derive implications of counterfactual invariance that can be measured in the observed data. Regularizing predictors to satisfy these observable implications provides a means for achieving (partial) counterfactual invariance. Then, we will connect counterfactual invariance to robust prediction under certain domain shifts, with the aim of clarifying what counterfactual invariance buys and when it is desirable.

An important insight that emerges from the formalization is that the true underlying causal structure of the data has fundamental implications for both model training and guarantees. Methods for handing 'spurious correlations' in data with a given causal structure need not perform well when blindly translated to another causal structure.

**Counterfactual Invariance** Consider the problem of learning a predictor $f$ that predicts a label $Y$ from covariates $X$. In this paper, we're interested in constructing predictors whose predictions are invariant to certain perturbations on $X$. Our first task is to formalize the invariance requirement.

To that end, assume that there is an additional variable $Z$ that captures information that should not influence predictions. However, $Z$ may causally influence the covariates $X$. Using the potential outcomes notation, let $X(z)$ to denote the counterfactual $X$ we would have seen had $Z$ been set to $z$, leaving all else fixed. Informally, we can understand perturbative stress tests as a way of producing particular realizations of counterfactual pairs $X(z)$, $X(z')$ that differ by an intervention on $z$. Then, we formalize the requirement that an arbitrary change to $z$ does not change predictions:

**Definition 1.1.** A predictor $f$ is *counterfactually invariant to* $Z$ if $f(X(z)) = f(X(z'))$ almost everywhere, for all $z, z'$ in the sample space of $Z$. When $Z$ is clear from context, we'll just say the predictor is counterfactually invariant.

## 2 Causal Structure

Counterfactual invariance is a condition on how the *predicted* label behaves under interventions on parts of the input data. However, intuitions about stress testing are based on how the *true* label behaves under interventions on parts of the input data. We will see that the true causal structure fundamentally affects both the implications of counterfactual invariance, and the techniques we use to achieve it. To study this phenomenon, we'll use two causal structures that are commonly encountered in applications; see Figure 1.

### 2.1 Prediction in the Causal Direction

We begin with the case where $X$ is a cause of $Y$.

**Example 2.1.** We want to automatically classify the quality of product reviews. Each review has a number of "helpful" votes $Y$ (from site users). We predict $Y$ using the text of the product review $X$. However, we find interventions on the sentiment $Z$ of the text change our prediction; changing "Great shoes!" to "Bad shoes!" changes the prediction.

In the examples in this paper, the covariate $X$ is text data. Usually, the causal relationship between the text and $Y$ and $Z$ will be complex— e.g., the relationships may depend on abstract, unlabeled, parts of the text such as topic, writing quality, or tone. In principle, we could enumerate all such latent variables, construct a causal graph capturing the relationships between these variables and $Y, Z$, and use this causal structure to study counterfactual invariance. For instance, if we think that topic causally influences the helpfulness $Y$, but is not influenced by sentiment $Z$, then we could build a counterfactually invariant predictor by extracting the topic and

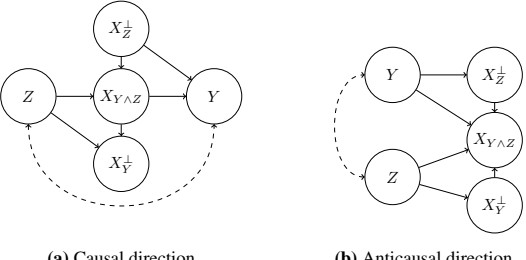

(a) Causal direction     (b) Anticausal direction

**Figure 1:** Causal models for the data generating process. We decompose the observed covariate $X$ into latent parts defined by their causal relationships with $Z$ and $Y$. Solid arrows denote causal relationships, while dashed lines denote non-causal associations. The differences between these causal structures will turn out to be key for understanding counterfactual invariance.

predicting $Y$ using topic alone. However, exhaustively articulating all possible such variables is a herculean task.

Instead, notice that the only thing that's relevant about these latent variables is their causal relationship with $Y$ and $Z$. Accordingly, we'll decompose the observed variable $X$ into parts defined by their causal relationships with $Y$ and $Z$. We remain agnostic to the semantic interpretation of these parts. Namely, we define $X_{\bar{Z}}^{\perp}$ as the part of $X$ that is not causally influenced by $Z$ (but may influence $Y$), $X_{\bar{Y}}^{\perp}$ as the part that does not causally influence $Y$ (but may be influenced by $Z$), and $X_{Y \wedge Z}$ is the remaining part that is both influenced by $Z$ and that influences $Y$. The causal structure is shown in Figure 1a.

We see there are two paths that lead to $Y$ and $Z$ being associated. The first is when $Z$ affects $X_{Y \wedge Z}$ which, in turn, affects $Y$. For example, a very enthusiastic reviewer might write a longer, more detailed review, which will in turn be more helpful. The second is when a common cause or selection effect in the data generating process induces an association between $Z$ and $Y$, which we denote with a dashed arrow. For example, if books tend to get more positive reviews, and also people who buy books are more likely to flag reviews as helpful, then the product type would be a common cause of sentiment and helpfulness.

## 2.2 Prediction in the Anti-Causal Direction

We also consider the case where $Y$ causes $X$.

**Example 2.2.** We want to predict the star rating $Y$ of movie reviews from the text $X$. However, we find that predictions are influenced by the movie genre $Z$; e.g., changing "Adam Sandler" (a comedy actor) to "Hugh Grant" (a romance actor) changes the predictions.

Figure 1b shows the causal structure. Here, the observed $X$ is influenced by both $Y$ and $Z$. Again, we decompose $X$ into parts defined by their causal relationship with $Z$ and $Y$. Here, $Z$ (and thus $X_{\bar{Y}}^{\perp}$) can be associated with $Y$ through two paths. First, if $X_{Y \wedge Z}$ is non-trivial, then conditioning on it causes a dependence between $Z$ and $Y$ (because $X_{Y \wedge Z}$ is a collider). For example, if Adam Sandler tends to appear in good comedy movies but bad movies of other genres then seeing "Sandler" in the text induces a dependency between sentiment and genre. Second, $Z$ and $Y$ may be associated due to a common cause, or due to selection effects in the data collection protocol—this is represented by the dashed line between $Z$ and $Y$. For example, fans of romantic comedies may tend to give higher reviews (to all films) than fans of horror movies.

## 2.3 Non-Causal Associations

Frequently, a predictor trained to predict $Y$ from $X$ will rely on $X_{\bar{Y}}^{\perp}$, *even though there is no causal connection between $Y$ and $X_{\bar{Y}}^{\perp}$*, and therefore will fail counterfactual invariance. The reason is that $X_{\bar{Y}}^{\perp}$ serves as a proxy for $Z$, and $Z$ is predictive of $Y$ due to the non-causal (dashed line) association.

There are two mechanisms that can induce such associations. First, $Y$ and $Z$ may be *confounded*: they are both influenced by an unobserved common cause $U$. For example, people who review books may be more upbeat than people who review clothing. This leads to positive sentiments and high helpfulness votes for books, creating an association between sentiment and helpfulness. Second, $Y$ and $Z$ may be subject to *selection*: there is some condition (event) $S$ that depends on $Y$ and $Z$, such that a data point from the population is included in the sample only if $S = 1$ holds. For example, our training data might only include movies with at least 100 reviews. If only excellent horror movies have so many reviews (but most rom-coms get that many), then this selection would induce an association between genre and score. Formally, the dashed-line causal graphs mean our sample is distributed according to $P(X, Y, Z) = \int P(X, Y, Z, u \mid S = 1) \mathrm{d}P(u)$ where $Y, Z$ are caused by $U$ and are causes of $S$, and $(X, Y, Z)$ are causally related according to the graph.

In addition to the non-causal dashed-line relationship, there is also dependency induced by between $Y$ and $Z$ by $X_{Y \wedge Z}$. Whether or not each of these dependencies is "spurious" is a problem-specific judgement that must be made by each analyst based on their particular use case. E.g., using genre to predict sentiment may or may not be reasonable, depending on the actual application in mind. However, there is a special case that captures a common intuition for purely spurious association.

**Definition 2.3.** We say that the association between $Y$ and $Z$ is *purely spurious* if $Y \perp\!\!\!\perp X \mid X_{\bar{Z}}^{\perp}, Z$.

That is, if the dashed-line association did not exist (removed by conditioning on $Z$) then the part of $X$ that is not influenced by $Z$ would suffice to estimate $Y$.

# 3 Observable Signatures of Counterfactually Invariant Predictors

We now consider the question of how to achieve counterfactual invariance in practice. The challenge is that counterfactual invariance is defined by the behavior of the predictor on counterfactual data that is never actually observed. This makes checking counterfactual invariance impossible. Instead, we'll derive a signature of counterfactual invariance that actually can be measured—and enforced—using ordinary datasets where $Z$ (or a proxy) is measured. For example, the star rating of a review as a proxy for sentiment, or genre labels in the movie review case.

Intuitively, a predictor $f$ is counterfactually invariant if it depends only on $X_Z^\perp$, the part of $X$ that is not affected by $Z$. To formalize this, we need to show that such a $X_Z^\perp$ is well defined:

**Lemma 3.1.** *Let $X_Z^\perp$ be a $X$-measurable random variable such that, for all measurable functions $f$, we have that $f$ is counterfactually invariant if and only if $f(X)$ is $X_Z^\perp$-measurable. If $Z$ is discrete[1] then such a $X_Z^\perp$ exists.*

Accordingly, we'd like to construct a predictor that is a function of $X_Z^\perp$ only (i.e., is $X_Z^\perp$ measurable). The key insight is that we can use the causal graphs to read off a set of conditional independence relationships that are satisfied by $Z, X_Z^\perp, Y$. Critically, these conditional independence relationships are testable from the observed data. Thus, they provide a signature of counterfactual invariance:

**Theorem 3.2.** *If $f$ is a counterfactually invariant predictor:*

1. *Under the anti-causal graph, $f(X) \perp\!\!\!\perp Z \mid Y$.*
2. *Under the causal-direction graph, if $Y$ and $Z$ are not subject to selection (but possibly confounded), $f(X) \perp\!\!\!\perp Z$.*
3. *Under the causal-direction graph, if the association is purely spurious, $Y \perp\!\!\!\perp X \mid X_Z^\perp, Z$, and $Y$ and $Z$ are not confounded (but possibly selected), $f(X) \perp\!\!\!\perp Z \mid Y$.*

*Remark* 3.3 (Connection to Fairness). This result has an interesting interpretation when $Z$ is a protected attribute (e.g., sex or race) that we'd like to be fair with respect to. There are many ways of formalizing fairness, which are usually mutually incompatible. In the fairness setting, counterfactual invariance is equivalent to counterfactual fairness [Kus+17; Gar+19], the condition $f(X) \perp\!\!\!\perp Z$ is equivalent to demographic parity, and the condition $f(X) \perp\!\!\!\perp Z \mid Y$ is equivalent to equalized odds [Meh+19]. Theorem 3.2 then says that counterfactual fairness implies either demographic parity or equalized odds, depending on the true underlying causal structure of the problem. Hence, the relationship between disparate fairness notions is clarified by taking the underlying causal structure into account. This also suggests we can take counterfactual fairness as the fundamental notion, then use demographic parity in the causal-confounding case and equalized odds otherwise. However, we leave the (ethical) question of whether this is a sound strategy to future work.

**Causal Regularization** Without access to counterfactual examples, we cannot directly enforce counterfactual invariance. However, we can require a trained model to satisfy the counterfactual invariance signature of Theorem 3.2. The hope is that enforcing the signature will lead the model to be counterfactually invariant. To do this, we regularize the model to satisfy the appropriate conditional independence condition. For simplicity of exposition, we restrict to binary $Y$ and $Z$. The (infinite data) regularization terms are

$$\text{marginal regularization} = \text{MMD}(\text{P}(f(X) \mid Z = 0), \text{P}(f(X) \mid Z = 1)) \qquad (3.1)$$

$$\text{conditional regularization} = \text{MMD}(\text{P}(f(X) \mid Z = 0, Y = 0), \text{P}(f(X) \mid Z = 1, Y = 0)) \quad (3.2)$$
$$+ \text{MMD}(\text{P}(f(X) \mid Z = 0, Y = 1), \text{P}(f(X) \mid Z = 1, Y = 1)).$$

Maximum mean discrepancy (MMD) is a metric on probability measures.[2] The marginal independence condition is equivalent to (3.1) equal 0, and the conditional independence is equivalent to (3.2)

---

[1]In fact, it suffices that all potential outcomes $\{Y(z)\}_z$ are jointly measurable with respect to a single well-behaved sigma algebra; discrete $Z$ is sufficient but not necessary.

[2]The choice of regularization by MMD is for concreteness. Any technique for enforcing the independence signatures would do in principle—e.g., adversarial methods borrowed from the fairness literature. The key point here is the observation that different causal structures imply different independence signatures.

equal 0. In practice, we can estimate the MMD with finite data samples [Gre+12]. When training with stochastic gradient descent, we compute the penalty on each minibatch.

The procedure is then: if the data has causal-direction structure and the $Y \leftrightarrow Z$ association is due to confounding, add the marginal regularization term to the the training objective. If the data has anti-causal structure, or the association is due to selection, add the conditional regularization term instead. In this way, we regularize towards models that satisfy the counterfactual invariance signature.

A key point is that the regularizer we must use depends on the true causal structure. The conditional and marginal independence conditions are generally incompatible. Enforcing the condition that is mismatched to the true underlying causal structure may fail to encourage counterfactual invariance, or may throw away more information than is required.

**Gap to Counterfactual Invariance**  The conditional independence signature of Theorem 3.2 is necessary but not sufficient for counterfactual invariance. This is for two reasons. First, counterfactual invariance applies to individual datapoint realizations, but the signature is distributional. In particular, the invariance $\mathrm{P}(f(X) \mid \mathrm{do}(Z = z)) = \mathrm{P}(f(X) \mid \mathrm{do}(Z = z'))$ for all $z, z'$ would also imply the conditional independence signature. But, this invariance is weaker than counterfactual invariance, since it doesn't require access to counterfactual realizations. Second, $f(X) \perp\!\!\!\perp Z$ does not imply, in general, that $Z$ is not a cause of $f(X)$. This (unusual) behavior can happen if, e.g., there are levels of $Z$ that we do not observe in the training data, or there are variables omitted from the causal graph that are a common cause of $Z$ and $X$.

Unfortunately, the gap between the signature and counterfactual invariance is a fundamental restriction of using observational data. The conditional independence signature is in some sense the closest proxy for counterfactual invariance we can hope for. In section 5, we'll see that enforcing the signature does a good job of enforcing counterfactual invariance in practice.

## 4 Performance Out of Domain

Counterfactual invariance is an intuitively desirable property for a predictor to have. However, it's not immediately clear how it relates to model performance as measured by, e.g., accuracy. Intuitively, eliminating predictor dependence on a spurious $Z$ may help with domain shift, where the data distribution in the target domain differs from the distribution of the training data. We now turn to formalizing this idea.

First, we must articulate the set of domain shifts to be considered. In our setting, the natural thing is to hold the causal relationships fixed across domains, but to allow the non-causal ("spurious") dependence between $Y$ and $Z$ to vary. Demanding that the causal relationships stay fixed reflects the requirement that the causal structure describes the dynamics of an underlying real-world process— e.g., the author's sentiment is always a cause (not an effect) of the text in all domains. On the other hand, the dependency between $Y$ and $Z$ induced by either confounding or selection can vary without changing the underlying causal structure. For confounding, the distribution of the confounder may differ between domains—e.g., books are rare in training, but common in deployment. For selection, the selection criterion may differ between domains—e.g., we include only frequently reviewed movies in training, but make predictions for all movies in deployment.

We want to capture spurious domain shifts by considering domain shifts induced by selection or confounding. However, there is an additional nuance. Changes to the marginal distribution of $Y$ will affect the risk of a predictor, even in the absence of any spurious association between $Y$ and $Z$. Therefore, we restrict to shifts that preserve the marginal distribution of $Y$.

**Definition 4.1.** We say that distributions $P, Q$ are *causally compatible* if both obey the same causal graph, $P(Y) = Q(Y)$, and there is a confounder $U$ and/or selection conditions $S, \tilde{S}$ such that $P = \int \mathrm{P}(X, Y, Z \mid U, S = 1) \mathrm{d}\tilde{P}(U)$ and $Q = \int \mathrm{P}(X, Y, Z \mid U, \tilde{S} = 1) \mathrm{d}\tilde{Q}(U)$ for some $\tilde{P}(U), \tilde{Q}(U)$.

We can now connect counterfactual invariance and robustness to domain shift.

**Theorem 4.2.** *Let $\mathcal{F}^{\mathrm{invar}}$ be the set of all counterfactually invariant predictors. Let $L$ be either square error or cross entropy loss. And, let $f^* := \mathrm{argmin}_{f \in \mathcal{F}^{\mathrm{invar}}} \mathbb{E}_P[L(Y, f(X))]$ be the counterfactually invariant risk minimizer. Suppose that the target distribution $Q$ is causally compatible with the training distribution $P$. Suppose that any of the following conditions hold:*

1. *the data obeys the anti-causal graph*
2. *the data obeys the causal-direction graph, there is no confounding (but possibly selection), and the association is purely spurious, $Y \perp\!\!\!\perp X \mid X_{\overline{Z}}^{\perp}, Z$, or*
3. *the data obeys the causal-direction graph, there is no selection (but possibly confounding), the association is purely spurious and the causal effect of $X_{\overline{Z}}^{\perp}$ on $Y$ is additive, i.e., the true data generating process is*

$$Y \leftarrow g(X_{\overline{Z}}^{\perp}) + \tilde{g}(U) + \xi \text{ where } \mathbb{E}[\xi \mid X_{\overline{Z}}^{\perp}] = 0, \tag{4.1}$$

*for some functions $g, \tilde{g}$.*

*Then, the training domain counterfactually invariant risk minimizer is also the target domain counterfactually invariant risk minimizer, $f^* = \operatorname{argmin}_{f \in \mathcal{F}^{\text{invar}}} \mathbb{E}_Q[L(Y, f(X))]$.*

*Remark* 4.3. The causal case with confounding requires an additional assumption (additive structure) because, e.g., an interaction between confounder and $X_{\overline{Z}}^{\perp}$ can yield a case where $X_{\overline{Z}}^{\perp}$ and $Y$ have a different relationship in each domain (whence, out-of-domain learning is impossible).

This result gives a recipe for finding a good predictor in the target domain even without access to any target domain examples at training time. Namely, find the counterfactually invariant risk minimizer in the training domain. In practice, we can use the regularization scheme of section 3 to (approximately) achieve this. We'll see in section 5 that this works well in practice.

**Optimality**  Theorem 4.2 begs the question: if the only thing we know about the target setting is that it's causally compatible with the training data, is the best predictor the counterfactually invariant predictor with lowest training risk? A natural way to formalize this question is to study the predictor with the best performance in the worst case target distribution. We define $\mathcal{Q} = \{Q : Q \text{ causally compatible with } P\}$ and the $\mathcal{Q}$-minimax predictor $f^*_{\text{minimax}} = \operatorname{argmin}_{f \in \mathcal{F}} \max_{Q \in \mathcal{Q}} \mathbb{E}_Q[L(Y, f(X))]$. The question is then: what's the relationship between the counterfactually invariant risk minimizer and the minimax predictor?

**Theorem 4.4.** *The counterfactually invariant risk minimizer is not $\mathcal{Q}$-minimax in general. However, under the conditions of Theorem 4.2, if the association is purely spurious, $X_{Y \wedge Z} \perp\!\!\!\perp Y \mid X_{\overline{Z}}^{\perp}, Z$, and $\mathrm{P}(Z, Y)$ satisfies overlap, then the two predictors are the same. By overlap we mean that $\mathrm{P}(Z, Y)$ is a discrete distribution such that for all $(z, y)$, if $\mathrm{P}(z, y) > 0$ then there is some $y' \neq y$ such that also $\mathrm{P}(z, y') > 0$.*

Conceptually, Theorem 4.4 just says that the counterfactually invariant predictor excludes $X_{Y \wedge Z}$, even when this information is useful in every domain. In the purely spurious case, $X_{Y \wedge Z}$ carries no useful information, so counterfactual invariance is optimal.

# 5   Experiments

The main claims of the paper are:

1. Stress test violations can be reduced by suitable conditional independence regularization.
2. This reduction will improve out-of-domain prediction performance.
3. To get the full effect, the imposed penalty must match the causal structure of the data.

**Setup**  To assess these claims, we'll examine the behavior of predictors trained with the marginal or conditinal regularization on multiple text datasets that have either causal or anti-causal structure. We expect to see that marginal regularization improves stress test and out-of-domain performance on data with causal-confounded structure, and conditional regularization improves these on data with anti-causal structure.

For each experiment, we use BERT [Dev+19] finetuned to predict a label $Y$ from the text as our base model. We train multiple causally-regularized models on the each dataset. The training varies by whether we use the conditional or marginal penalty, and by the strength of the regularization term. That is, we train identical architectures using $\mathrm{CrossEntropy} + \lambda \cdot \mathrm{Regularizer}$ as the objective function, where we vary $\lambda$ and take $\mathrm{Regularizer}$ as either the marginal penalty, (3.1), or conditional penalty, (3.2). We compare these models' predictions on data with causal and anti-causal structure.

See supplement for experimental details.

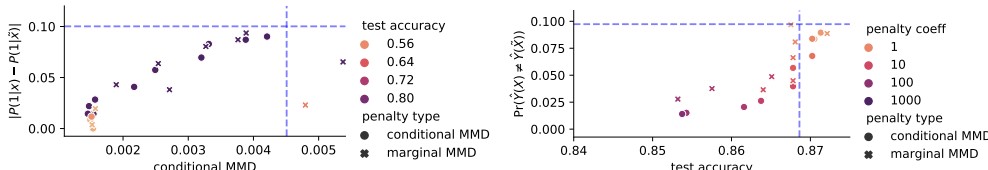

**Figure 2: Regularizing conditional MMD improves counterfactual invariance on synthetic anti-causal data**. Sufficiently high regularization of marginal MMD also improves invariance, but impairs accuracy. Dashed lines show baseline performance of an unregularized predictor. **Left**: lower conditional MMD implies that predictive probabilities are invariant to perturbation. Although marginal MMD penalization can result in low conditional MMD and good stress test performance, this comes at the cost of very low in-domain accuracy. **Right**: MMD regularization reduces the rate of predicted label flips on perturbed data, with little affect on in-domain accuracy. Conditional MMD regularization reduces predicted label flips to $1.4\%$, while the best result for marginal MMD is $2.8\%$.

## 5.1 Robustness to Stress Tests

First, we examine whether enforcing the causal regularization actually helps to enforce counterfactual invariance. We create counterfactual (stress test) examples by perturbing the input data and compare the prediction on these. We build the experimental datasets using Amazon reviews from the product category "Clothing, Shoes, and Jewelry" [NLM19].

**Synthetic**  To study the relationship between counterfactual invariance and the distributional signature of Theorem 3.2, we construct a synthetic confound. For each review, we draw a Bernoulli random $Z$, and then perturb the text $X$ so that the common words "the" and "a" carry information about $Z$: for example, we replace "the" with the token "thexxxxx" when $Z = 1$. We take $Y$ to be the review score, and subsample so $Y$ is balanced. This data has anti-causal structure: the text $X$ is written to explain the score $Y$. Further, we expect that the $Y, Z$ association is purely spurious, because "the" and "a" carry little information about the label.

We train the models on data where $\mathrm{P}(Y = Z) = 0.3$. We then create perturbed stress-test datasets by changing each example $X_i(z)$ to the counterfactual $X_i(1 - z)$ (using the synthetic model). By measuring the performance of each model on the perturbed data, we can test whether the distributional properties enforced by the regularizers result in counterfactual invariance at the instance level. Figure 2 shows that conditional regularization (matching the anti-causal structure) reduces checklist failures, as measured by the frequency that the predicted label changes due to perturbation as well as the mean absolute difference in predictive probabilities that is induced by perturbation.

**Natural**  To study the relationship in real data, we use the review data in a different way. We now take $Z$ to be the score, binarized as $Z \in \{1 \text{ or } 2 \text{ stars}, 4 \text{ or } 5 \text{ stars}\}$. We use this $Z$ as a proxy for sentiment, and consider problems where sentiment should (plausibly) not have a causal effect on $Y$. For the causal prediction problem, we take $Y$ to be the helpfulness score of the review (binarized as described below). This is causal because readers decide whether the review is helpful based on the text. For the anti-causal prediction problem, we take $Y$ to be whether "Clothing" is included as a category tag for the product under review (e.g., boots typically do not have this tag). This is anti-causal because the product category affects the text.

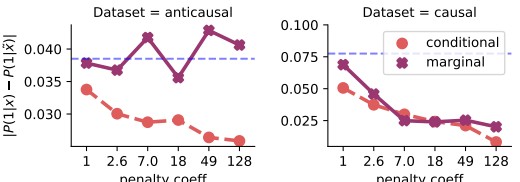

**Figure 3: Penalizing the MMD matching the causal structure improves stress test performance** on natural product review data. Note that penalizing the wrong MMD may not help: the marginal MMD hurts on the anticausal dataset. Perturbations are generated by swapping positive and negative sentiment adjectives in examples.

We control the strength of the spurious association between $Y$ and $Z$. In the anti-causal case, this is done by selection: we randomly subset the data to enforce a target level of dependence between $Y$ and $Z$. The causal-direction case with confounding is more complicated. To manipulate confounding strength, we binarize the number of helpfulness votes $V$ in a manner determined by

the target level of association. We take $Y = 1[V > T_Z]$ where $T_Z$ is a $Z$-dependent threshold, chosen to induce a target association. We choose $P(Y = 0 \mid Z = 0) = P(Y = 1 \mid Z = 1) = 0.3$. We balance $Z$ by subsampling, which also balances $Y$.

Now, we create stress test perturbations of these datasets by randomly changing adjectives in the examples. Using predefined lists of postive sentiment adjectives and negative sentiment adjectives, we swap any adjective that shows up on a list with a randomly sampled adjective from the other list. This preserves basic sentence structure, and thus creates a limited set of counterfactual pairs that differ on sentiment.

Results for differences in predicted probabilities between original and perturbed data are shown in Figure 3. Each point is a trained model, which vary in measured MMD on the test data and on sensitivity to perturbations. Recall that the conditional independence signature of Theorem 3.2 are necessary but not sufficient for counterfactual invariance, so it's not certain that regularizing to reduce the MMD will reduce perturbation sensitivity. Happily, we see that regularizing to reduce the MMD that matches the causal structure does indeed reduce sensitivity to perturbations.

Notice that regularizing the causally mismatched MMD can have strange effects. Regularizing marginal MMD in the anti-causal case actually makes the model more sensitive to perturbations!

## 5.2 Domain Shift

Next, we study the effect of causal regularization on model performance under domain shift.

**Natural Product Review** We again use the natural Amazon review data described above. For both the causal and anti-causal data, we create multiple test sets with variable spurious correlation strength. This is done in the manner described above, varying $P(Y = 0 \mid Z = 0) = P(Y = 1 \mid Z = 1) = \gamma$. Here, $\gamma$ is the strength of spurious association. The test sets are out-of-domain samples. By design, $Y$ is balanced in each dataset, so these samples are causally compatible with the training data. For both the causal and anti-causal datasets, the training data has $P(Y = 0 \mid Z = 0) = P(Y = 1 \mid Z = 1) = 0.3$. We train a classifier for each regularization type and regularization strength, and measure the accuracy on each test domain. The results are shown in Figure 4.

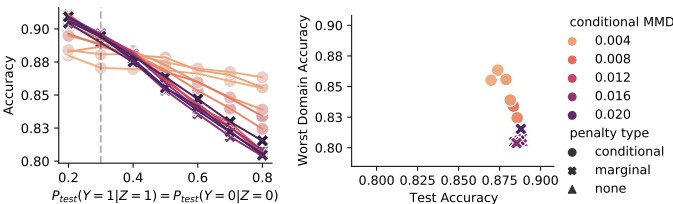

**Anti-Causal Data**: conditional regularization improves domain-shift robustness.

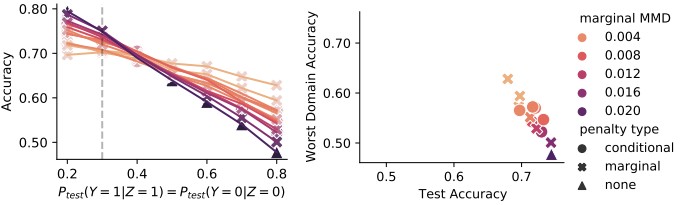

**Causal-Direction Data**: marginal regularization improves domain-shift robustness.

**Figure 4: The best domain-shift robustness is obtained by using the regularizer that matches the underlying causal structure of the data.** The plots show out-of-domain accuracy for models trained on the (natural) review data. In each row, the **left** figure shows out-of-domain accuracies (lines are models), with the $X$-axis showing the level of spurious correlation in the test data (0.3 is the training condition); the **right** figure shows worst out-of-domain accuracy versus in-domain test accuracy (dots are models).

First, the unregularized predictors do indeed learn to rely on the spurious association between sentiment and the label. The accuracy of these predictors decays dramatically as the spurious assocation moves from negative (0.3) to positive—in the causal case, the unregularized predictor is worse than chance in the 0.8 domain.

Following section 3, the regularization that matches the underlying causal structure should yield a predictor that is (approximately) counterfactually invariant. Following Theorem 4.2, we expect that good performance of a counterfactually-invariant predictor in the training domain should imply good performance in each of the other domains. Indeed, we see that this is so. Models that are regularized to have small values of the appropriate MMD do indeed have better out-of-domain performance.

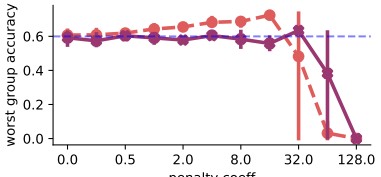 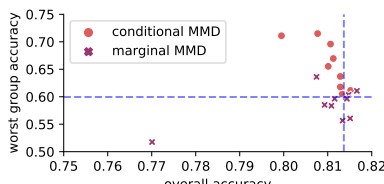

**Figure 5: Conditional MMD penalization improves robustness in anti-causal MNLI data.** Marginal regularization does not improve over the baseline unregularized model, shown with dashed lines. **Left:** Conditional regularization improves minimum accuracy across $(Y, Z)$ groups. When overregularized, the predictor returns the same $\hat{Y}$ for all inputs, yielding a worst-group accuracy of 0. **Right:** Conditional MMD regularization significantly improves worst $(Y, Z)$ group accuracy ($y$-axis) while only mildly reducing overall accuracy ($x$-axis).

Such models have somewhat worse in-domain performance, because they no longer exploit the spurious correlation.

**MNLI Data**  For an additional test on naturally-occurring confounds, we use the multi-genre natural language inference (MNLI) dataset [WNB18]. Instances are concatenations of two sentences, and the label describes the semantic relationship between them, $Y \in$ {contradiction, entailment, neutral}. There is a well-known confound in this dataset: examples where the second sentence contain a negation word (e.g., "not") are much more likely to be labeled as contradictions [Gur+18]. Following Sagawa et al. [Sag+20], we set $Z$ to indicate whether one of a small set of negation words is present. Although $Z$ is derived from the text $X$, it can be viewed as a proxy for a latent variable indicating whether the author intended to use negation in the text. This is an anti-causal prediction problem: the annotators were instructed to write text to reflect the desired label [WNB18].

Following Sagawa et al. [Sag+20], we divide the MNLI data into groups by $(Y, Z)$ and compute the "worst group accuracy" across all such groups. Because this is an anti-causal problem, we predict that the conditional MMD is a more appropriate penalty than the marginal MMD. As shown in Figure 5, this prediction holds: conditional MMD regularization dramatically improves performance on the worst group, while only lightly impacting the overall accuracy across groups.

## 6    Related work

Several papers draw a connection between causality and domain shifts [SS18; SCS19; Arj+20; Mei18; PBM16; RC+18; Zha+13; Sch+12]. Typically, this work considers a prediction setting where the covariates $X$ include both causes and effects of $Y$, and it is unknown which is which. The goal is to learn to predict $Y$ using only its causal parents. Zhang et al. [Zha+13] considers anti-causal domain shift induced by changing $P(Y)$ and proposes a data reweighting scheme. By contrast, counterfactual invariance is an example-wise notion of invariance, not an invariance across environments. In particular, learning a counterfactually invariant predictor only requires access to data from a single environment. Theorem 4.2 establishes that counterfactual invariance does imply a certain domain generalization guarantee over causally-compatible domains. Note though that the notion of causal compatibility is not generally the same as class of domain shifts previously considered. For example, we have invariant performance in the anti-causal setting, but this is ruled out by Arjovsky et al. [Arj+20].

A related body of work focuses on "causal representation learning" [Bes+19; Loc+20; Sch+21; Arj+20]. Our approach follows this tradition, but focuses on splitting $X$ into components defined by their causal relationships with the label $Y$ and an additional covariate $Z$. Rather than attempting to infer the causal relationship between $X$ and $Y$, we show that domain knowledge of this relationship is essential for obtaining counterfactually-invariant predictors. The role of causal vs anti-causal data generation in semi-supervised learning and transfer learning has also been studied [Sch+21; Sch+12]. In this paper we focus on a different implication of the causal vs anti-causal distinction.

Another line of work considers the case where either the counterfactuals $X(z)$, $X(z')$ are observed for at least some data points, or where it's assumed that there's enough structural knowledge to (learn to) generate counterfactual examples [RPH21; Wu+21; Gar+19; Mit+20; WZ19; KCC20; KHL20; TAH20]. Kusner et al. [Kus+17] and Garg et al. [Gar+19] in particular examine a notion of

counterfactual fairness that can be seen as equivalent to counterfactual invariance. In these papers, approximate counterfactuals are produced by direct manipulation of the features (e.g., change male to female names), generative models (e.g., style transfer of images), or crowdsourcing. Then, these counterfactuals can either be used as additional training data or the predictor can be regularized such that it cannot distinguish between $X_i(z)$ and $X_i(z')$. This strategy can be viewed as enforcing counterfactual invariance directly; an advantage is that it avoids the necessary-but-not-sufficient nuance of Theorem 3.2. However, counterfactual examples can be difficult to obtain for language data in many realistic problem domains, and it may be difficult to learn to generalize from such examples [HLB20].

The marginal and conditional independencies of Theorem 3.2 have appeared in other contexts. As discussed in remark 3.3, if we think of $Z$ as a protected attribute and $f$ as a 'fair' classifier, then counterfactual invariance is counterfactual fairness, the marginal independence is demographic parity, and the conditional independence is equalized odds [Meh+19]. In another setting, one approach to domain adaptation is to seek representations $\phi$ such that either $\phi(X)$ [e.g., MBS13; Bak+13; Gan+16; Tze+14] or $\phi(X) \mid Y$ [e.g., MLM19; Yan+17] have the same distribution in each environment. The connection here is subtler; it is inappropriate to interpret $Z$ as a domain label (we only have access to one domain at train time, and all values of $Z$ are present in each domain). To clarify the connection, consider introducing an extra variable $E$ that labels the domain. For concreteness, consider the anti-causal case where the spurious association between $Y$ and $Z$ is due to an observed confounder $U$. Now, suppose that $E$ is a cause of $U$. Then, $E$ labels the distribution of the unobserved confounder, and thus different values of $E$ correspond to different causally compatible domains. Now, if we draw the causal graph we can read off that $X_z^\perp E|Y$. That is, we arrive at the same conditional independence criterion. However, it's essentially a coincidence that this matches the criterion we'd use if we'd observed $Z$—the two variables have totally different causal meanings and interpretations. For example, in the review data case, we might take $Z$ to be review score and $E$ to be the website the review is collected from. The review quality doesn't need to be counterfactually invariant to the website, and indeed we wouldn't expect it to be!

Finally, this work can be viewed as part of a recent line of work using counterfactuals to examine the connection between example-wise robustness (stress testing) and distributional-level robustness (domain shift) [e.g., TAH20; RPH21; Kau+21]. Teney et al. [TAH20] use near counterfactuals with different labels in a modified training objective, and show improved performance out-of-domain empirically. Robey et al. [RPH21] use image models to generate counterfactuals, and incorporate the model into training, showing out-of-domain improvement. They assume a somewhat different causal structure—articulating the precise connection is an interesting future direction. Kaushik et al. [Kau+21] study a linear-Gaussian model and argue that out-of-domain performance can be used as a signal to distinguish 'causal' and 'spurious' features.

# 7 Discussion

We used the tools of causal inference to formalize and study perturbative stress tests. A main insight of the paper is that counterfactual desiderata can be linked to observationally-testable conditional independence criteria. This requires consideration of the true underlying causal structure of the data. Done correctly, the link yields a simple procedure for enforcing the counterfactual desiderata, and mitigating the effects of domain shift.

The main limitation of the paper is the restrictive causal structures we consider. In particular, we require that $X_{\bar{Z}}^\perp$, the part of $X$ not causally affected by $Z$, is also statistically independent of $Z$ in the observed data. However, in practice these may be dependent due to a common cause. In this case, the procedure here will be overly conservative, throwing away more information than required. Additionally, it is not obvious how to apply the ideas described here to more complicated causal situations, which can occur in structured prediction (e.g., question answering). Extending the ideas to handle richer causal structures is an important direction for future work. The work described here can provide a template for this research program.

# 8 Acknowledgements

Thanks to Kevin Murphy, Been Kim, Molly Roberts, Justin Grimmer, and Brandon Stewart for feedback on an earlier version.

# 9 Funding Transparency

This paper was not supported by any third party funding, and the authors have no competing interests to declare.

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
