# A  Proofs

**Lemma 3.1.** *Let $X_Z^\perp$ be a $X$-measurable random variable such that, for all measurable functions $f$, we have that $f$ is counterfactually invariant if and only if $f(X)$ is $X_Z^\perp$-measurable. If $Z$ is discrete[3] then such a $X_Z^\perp$ exists.*

*Proof.* Write $\{X(z)\}_z$ for the potential outcomes. First notice that if $f(X)$ is $\{X(z)\}_z$-measurable then $f(X)$ is counterfactually invariant. This is essentially by definition—intervention on $Z$ doesn't change the potential outcomes, so it doesn't change the value of $f(X)$. Conversely, if $f$ is counterfactually invariant, then $f(X)$ is $\{X(z)\}_z$-measurable. To see this, notice that $X = \sum_z 1[Z = z]X(z)$ is determined by $Z$ and $\{X(z)\}_z$, so $f(X) = \tilde{f}(Z, \{X(z)\}_z)$ for $\tilde{f}(z, \{x(z)\}_z) = f(\sum_z' 1[z' = z]x(z))$. Now, if $\tilde{f}$ depends only on $\{X(z)\}_z$ we're done. So suppose that there is $z, z'$ such that $\tilde{f}(z, \{X(z)\}_z) \neq \tilde{f}(z', \{X(z)\}_z)$ (almost everywhere). But then $f(X(z)) \neq f(X(z'))$, contradicting counterfactual invariance.

Now, define $\mathcal{F}_{X_Z^\perp} = \sigma(X) \wedge \sigma(\{X(z)\}_z)$ as the intersection of sigma algebra of $X$ and the sigma algebra of the potential outcomes $\{X(z)\}_z$. Because $\mathcal{F}_{X_Z^\perp}$ is the intersection of sigma algebras, it is itself a sigma algebra. Because every $\mathcal{F}_{X_Z^\perp}$-measurable random variable is $\{X(z)\}_z$-measurable, we have that $Z$ is not a cause of any $\mathcal{F}_{X_Z^\perp}$-measurable random variable (i.e., there is no arrow from $Z$ to $X_Z^\perp$). Because, for $f$ counterfactually invariant, $f(X)$ is both $X$-measurable and $\{X(z)\}_z$-measurable, it is also $\mathcal{F}_{X_Z^\perp}$-measurable. $\mathcal{F}_{X_Z^\perp}$ is countably generated, as $\{X(z)\}_z$ and $X$ are both Borel measurable. Therefore, we can take $X_Z^\perp$ to be any random variable such that $\sigma(X_Z^\perp) = \mathcal{F}_{X_Z^\perp}$. $\qquad\square$

**Theorem 3.2.** *If $f$ is a counterfactually invariant predictor:*

1. *Under the anti-causal graph, $f(X) \perp\!\!\!\perp Z \mid Y$.*
2. *Under the causal-direction graph, if $Y$ and $Z$ are not subject to selection (but possibly confounded), $f(X) \perp\!\!\!\perp Z$.*
3. *Under the causal-direction graph, if the association is purely spurious, $Y \perp\!\!\!\perp X \mid X_Z^\perp, Z$, and $Y$ and $Z$ are not confounded (but possibly selected), $f(X) \perp\!\!\!\perp Z \mid Y$.*

*Proof.* Reading $d$-separation from the causal graphs, we have $X_Z^\perp \perp\!\!\!\perp Z$ in the causal-direction graph when $Y$ and $Z$ are not selected on, and $X_Z^\perp \perp\!\!\!\perp Z \mid Y$ for the other cases. By assumption, $f$ is a counterfactually-invariant predictor, which means that $f$ is $X_Z^\perp$-measurable.

$\qquad\square$

**Theorem 4.2.** *Let $\mathcal{F}^{\mathrm{invar}}$ be the set of all counterfactually invariant predictors. Let $L$ be either square error or cross entropy loss. And, let $f^* := \mathrm{argmin}_{f \in \mathcal{F}^{\mathrm{invar}}} \mathbb{E}_P[L(Y, f(X))]$ be the counterfactually invariant risk minimizer. Suppose that the target distribution $Q$ is causally compatible with the training distribution $P$. Suppose that any of the following conditions hold:*

1. *the data obeys the anti-causal graph*
2. *the data obeys the causal-direction graph, there is no confounding (but possibly selection), and the association is purely spurious, $Y \perp\!\!\!\perp X \mid X_Z^\perp, Z$, or*
3. *the data obeys the causal-direction graph, there is no selection (but possibly confounding), the association is purely spurious and the causal effect of $X_Z^\perp$ on $Y$ is additive, i.e., the true data generating process is*

$$Y \leftarrow g(X_Z^\perp) + \tilde{g}(U) + \xi \text{ where } \mathbb{E}[\xi \mid X_Z^\perp] = 0, \tag{4.1}$$

   *for some functions $g, \tilde{g}$.*

*Then, the training domain counterfactually invariant risk minimizer is also the target domain counterfactually invariant risk minimizer, $f^* = \mathrm{argmin}_{f \in \mathcal{F}^{\mathrm{invar}}} \mathbb{E}_Q[L(Y, f(X))]$.*

---

[3]In fact, it suffices that all potential outcomes $\{Y(z)\}_z$ are jointly measurable with respect to a single well-behaved sigma algebra; discrete $Z$ is sufficient but not necessary.

*Proof.* First, since counterfactual invariance implies $X_Z^\perp$-measurable,

$$\operatorname*{argmin}_{f \in \mathcal{F}^{\mathrm{invar}}} \mathbb{E}_P[L(Y, f(X)] = \operatorname*{argmin}_{f} \mathbb{E}_P[L(Y, f(X_Z^\perp)].\tag{A.1}$$

It is well-known that under squared error or cross entropy loss the minimizer is $f^*(x_Z^\perp) = \mathbb{E}_P[Y \mid x_Z^\perp]$. By the same argument, the counterfactually invariant risk minimizer in the target domain is $\mathbb{E}_Q[Y \mid x_Z^\perp]$. Thus, our task is to show $\mathbb{E}_P[Y \mid x_Z^\perp] = \mathbb{E}_Q[Y \mid x_Z^\perp]$.

We begin with the anti-causal case. We have that $P(Y \mid X_Z^\perp) = P(X_Z^\perp \mid Y)P(Y)/\int P(X_Z^\perp \mid Y)\mathrm{d}P(Y)$. By assumption, $P(Y) = Q(Y)$. So, it suffices to show that $P(X_Z^\perp \mid Y) = Q(X_Z^\perp \mid Y)$. To that end, from the anti-causal direction graph we have that $X_Z^\perp \perp\!\!\!\perp S, U \mid Y$. Then,

$$P(X_Z^\perp \mid Y) = \int \mathrm{P}(X_Z^\perp \mid Y, U, S = 1)\mathrm{d}\tilde{P}(U)\tag{A.2}$$

$$= \int \mathrm{P}(X_Z^\perp \mid Y, U, \tilde{S} = 1)\mathrm{d}\tilde{Q}(U)\tag{A.3}$$

$$= Q(X_Z^\perp \mid Y),\tag{A.4}$$

where the first and third lines are causal compatibility, and the second line is from $X_Z^\perp \perp\!\!\!\perp S, \tilde{S}, U \mid Y$.

The causal-direction case with no confounding follows essentially the same argument.

For the causal-direction case without selection,

$$\mathbb{E}_P[Y \mid X_Z^\perp] = g(X_Z^\perp) + \mathbb{E}_P[\tilde{g}(U) \mid X_Z^\perp] + \mathbb{E}_P[\xi \mid X_Z^\perp]\tag{A.5}$$

$$= g(X_Z^\perp) + \mathbb{E}_P[\tilde{g}(U)] + 0.\tag{A.6}$$

The first line is the assumed additivity. The second line follows because $\mathbb{E}_P[\xi \mid X_Z^\perp] = 0$ for all causally compatible distributions ($\mathrm{P}(\xi, X_Z^\perp)$ doesn't change), and $U \perp\!\!\!\perp X_Z^\perp$. Taking an expectation over $X_Z^\perp$, we have $\mathbb{E}_P[Y] = \mathbb{E}_P[g(X_Z^\perp)] + \mathbb{E}_P[\tilde{g}(U)]$. By the same token, $\mathbb{E}_Q[Y] = \mathbb{E}_Q[g(X_Z^\perp)] + \mathbb{E}_Q[\tilde{g}(U)]$. But, $\mathbb{E}_P[g(X_Z^\perp)] = \mathbb{E}_Q[g(X_Z^\perp)]$, since changes to the confounder don't change the distribution of $X_Z^\perp$ (that is, $X_Z^\perp \perp\!\!\!\perp U$). And, by assumption, $\mathbb{E}_Q[Y] = \mathbb{E}_P[Y]$. Together, these imply that $\mathbb{E}_P[\tilde{g}(U)] = \mathbb{E}_Q[\tilde{g}(U)]$. Whence, from (A.6), we have $\mathbb{E}_P[Y \mid X_Z^\perp] = \mathbb{E}_Q[Y \mid X_Z^\perp]$, as required. $\square$

**Theorem 4.4.** *The counterfactually invariant risk minimizer is not $\mathcal{Q}$-minimax in general. However, under the conditions of Theorem 4.2, if the association is purely spurious, $X_{Y \wedge Z} \perp\!\!\!\perp Y \mid X_Z^\perp, Z$, and $\mathrm{P}(Z, Y)$ satisfies overlap, then the two predictors are the same. By overlap we mean that $\mathrm{P}(Z, Y)$ is a discrete distribution such that for all $(z, y)$, if $\mathrm{P}(z, y) > 0$ then there is some $y' \neq y$ such that also $\mathrm{P}(z, y') > 0$.*

*Proof.* The reason that the predictors are not the same in general is that the counterfactually invariant predictor will always exclude information in $X_{Y \wedge Z}$, even when this information is helpful for predicting $Y$ in all target settings. For example, consider the case where $Y, Z$ are binary, $X = X_{Y \wedge Z}$ and, in the anti-causal direction, $X_{Y \wedge Z} = \mathrm{AND}(Y, Z)$. With cross-entropy loss, the counterfactually invariant predictor is just the constant $\mathbb{E}[Y]$, but the decision rule that uses $f(X) = 1$ if $X = 1$ is always better. In the causal case, consider $X_{Y \wedge Z} = Z$ and $Y = X_{Y \wedge Z}$.

Informally, the second claim follows because—in the absence of $X_{Y \wedge Z}$ information—any predictor $f$ that's better than the counterfactually invariant predictor when $Y$ and $Z$ are positively correlated will be worse when $Y$ and $Z$ are negatively correlated.

To formalize this, we begin by considering the case where $Y$ is binary and $X = X_Y^\perp$. So, in particular, the counterfactually invariant predictor is just some constant $c$. Let $f$ be any predictor that uses the information in $X_Y^\perp$. Our goal is to show that $\mathbb{E}_Q[L(f(X_Y^\perp), Y)] > \mathbb{E}_Q[L(c, Y)]$ for at least one test distribution (so that $f$ is not minimax). To that end, let $P$ be any distribution where $f(X_Y^\perp)$ has lower risk than $c$ (this must exist, or we're done). Then, define $A = \{(z, y) : \mathbb{E}_P[L(f(X_Y^\perp), y) \mid z] < L(c, y)\}$. In words: $A$ is the collection of $z, y$ points where $f$ did better than

the constant predictor. Since $f$ is better than the constant predictor overall, we must have $P(A) > 0$. Now, define $A^c = \{(z, 1 - y) : (z, y) \in A\}$. That is, the set constructed by flipping the label for every instance where $f$ did better. By the overlap assumption, $P(A^c) > 0$. By construction, $f$ is worse than $c$ on $A^c$. Further, $S = 1_A$ is a random variable that has the causal structure required by a selection variable (it's a child of $Y$ and $Z$ and nothing else). So, the distribution $Q$ defined by selection on $S$ is causally compatible with $P$ and satisfies $\mathbb{E}_Q[L(f(X_Y^\perp), Y)] > \mathbb{E}_Q[L(c, Y)]$, as required.

To relax the requirement that $X = X_Y^\perp$, just repeat the same argument conditional on each value of $X_Z^\perp$. To relax the condition that $Y$ is binary, swap the flipped label $1 - y$ for any label $y'$ with worse risk. □

# B   Experimental Details

## B.1   Model

All experiments use BERT as the base predictor. We use `bert_en_uncased_L-12_H-768_A-12` from TensorFlow Hub and do not modify any parameters. Following standard practice, predictions are made using a linear map from the representation layer. We use CrossEntropy loss as the training objective. We train with vanilla stochastic gradient descent, batch size 1024, and learning rate $1e - 5 \times 1024$. We use patience 10 early stopping on validation risk. Each model was trained using 2 Tensor Processing Units.

For the MMD regularizer, we use the estimator of Gretton et al. [Gre+12] with the Gaussian RBF kernel. We set kernel bandwidth to 10.0. We compute the MMD on $(\log f_0(x), \ldots, \log f_k(x))$, where $f_j(x)$ is the model estimate of $P(Y = k \mid x)$. (Note: this is log, not logit—the later has an extra, irrelevant, degree of freedom). We use log-spaced regularization coefficients between 0 and 128.

## B.2   Data

We don't do any pre-processing on the MNLI data.

The Amazon review data is from [NLM19].

### B.2.1   Inducing Dependence Between $Y$ and $Z$ in Amazon Product Reviews

To produce the causal data with $P(Y = 1 \mid Z = 1) = P(Y = 0 \mid Z = 0) = \gamma$

1. Randomly drop reviews with 0 helpful votes $V$, until both $P(V > 0 \mid Z = 1) > \gamma$ and $P(V > 0 \mid Z = 0) > 1 - \gamma$.
2. Find the smallest $T_z$ such that $P(V > T_1 \mid Z = 1) < \gamma$ and $P(V > T_0 \mid Z = 0) < 1 - \gamma$.
3. Set $Y = 1[V > T_0]$ for each $Z = 0$ example and $Y = 1[V > T_1]$ for each $Z = 1$ example.
4. Randomly flip $Y = 0$ to $Y = 1$ in examples where $(Z = 0, V = T_0 + 1)$ or $(Z = 1, V = T_1 + 1)$, until $P(Y = 1 \mid Z = 1) > \gamma$ and $P(Y = 1 \mid Z = 0) > 1 - \gamma$.

After data splitting, we have 58393 training examples, 16221 test examples, and 6489 validation examples.

To produce the anti-causal data with $P(Y = 1 \mid Z = 1) = P(Y = 0 \mid Z = 0) = \gamma$, choose a random subset with the target association. After data splitting, we have 157616 training examples, 43783 test examples, and 17513 validation examples.

### B.2.2   Synthetic Counterfactuals in Product Review Data

We select $10^5$ product reviews from the Amazon "clothing, shoes, and jewelery" dataset, and assign $Y = 1$ if the review is 4 or 5 stars, and $Y = 0$ otherwise. For each review, we use only the first twenty tokens of text. We then assign $Z$ as a Bernoulli random variable with $P(Z = 1) = \frac{1}{2}$. When $Z = 1$, we replace the tokens "and" and "the" with "andxxxxx" and "thexxxxx" respectively; for $Z = 0$ we use the suffix "yyyyy" instead. Counterfactuals can then be produced by swapping the suffixes. To induce a dependency between $Y$ and $Z$, we randomly resample so as to achieve $\gamma = 0.3$

and $P(Y = 1) = \frac{1}{2}$, using the same procedure that was used on the anti-causal model of "natural" product reviews. After selection there are $13,315$ training instances and $3,699$ test instances.