# OpenReview forum: "Counterfactual Invariance to Spurious Correlations in Text Classification"
_NeurIPS.cc/2021/Conference — NeurIPS 2021 Spotlight_

### Official Review · Reviewer_U7gK · 2021-07-16

**Rating:** 8
**Confidence:** 5

**Summary:**

In this paper, the authors study counterfactual invariance in machine learning models---intervening on “non causal” parts of the input should not change model predictions---and its relation to how models generalize out-of-domain. The authors show that counterfactually invariant predictors rely more on “causal” features as opposed to spurious correlations, thus generalizing to unseen distributions where these spurious correlations may not hold. The authors further propose two regularization schemes (one if the underlying data generation mechanism is represented by a causal model and another if it is an anticausal model) using which we can train counterfactually invariant predictors without access to counterfactual examples---as opposed to most prior work in which researchers often seek to either automatically construct or crowdsource counterfactual examples. They show that these strategies are heavily dependent on the underlying causal mechanisms, and that applying the regularization scheme designed for a causal model could lead to worse performance if the underlying data generation mechanism was anticausal and vice versa. The authors offer empirical evidence on text classification tasks (review helpfulness classification and natural language inference) to support their theoretical findings. I have some concerns about the paper (as discussed below) but overall I believe this paper builds on an exciting line of work and offers good theoretical and empirical contributions.

**Limitations And Societal Impact:**

I believe that the authors have provided a sound discussion of the limitations of proposed regularization schemes. But as they also showed in their experiments, regularizing causally mismatched MMD could make the model more reliant on spurious patterns, thus I would like the authors to include a discussion of how this limitation could potentially lead to catastrophic failures in critical applications.

**Main Review:**

- The paper is well motivated, and studies an important question of how predictive models may rely on causally relevant features or spurious correlations and the impacts on model performance out of domain.
- This paper is a significant contribution to an emerging line of research. To the best of my knowledge, so far there is only one paper that offers a theoretical explanation for the relationship between counterfactual invariance and out-of-domain generalization [1]. This paper takes a different route but arrives at the same conclusion, advancing our theoretical understanding of counterfactual invariance.
- There are some gaps between the conditional independence signature proposed in the paper and counterfactual invariance, however, to the authors’ credit they acknowledge that as a limitation of their methodology. However, this is not limited to their work, in fact, this as they correctly point out, this is a limitation of working with observational data.
- The proposed regularization schemes are dependent on the underlying causal mechanism (causal vs anticausal). From a practical point of view, requiring one to have knowledge of the underlying causal direction between the covariates and the label of interest can be a limiting factor in practical utility of proposed regularization schemes, as a mismatch between the enforced independence condition and underlying causal structure can lead to worse out of domain performance.
- The paper does not offer a discussion of prior work that provides an explanation for the effectiveness of counterfactual invariance as it relates to out-of-domain generalization. I believe this can be easily fixed in a future version. The paper could in general use a better discussion of related work, especially as it relates to Peters et al. (2016).
- The empirical investigation also has some limitations: first, I think the experiments section could use some background on dataset selection, processing, etc. Second, the whole focus of this paper is on how counterfactually invariant predictors generalize better to out of domain data. However, despite the presence of multiple datasets (representing plausible distribution shifts) for the tasks of how helpful a review is---check Yelp dataset, you can construct various test sets (businesses, restaurants, etc.)---and for natural language inference, the experiments are restricted only to Amazon reviews (from one category) in the former and MNLI in the latter. These results would be more convincing if the authors show that these results (better ood performance) hold on a battery of out-of-domain datasets.

I'm happy to improve my score if the authors can address these concerns.

There’s a typo on Line 252: “conditinal” → “conditional”

Curiosity question: Did you try experiments with regularizing the causally mismatched MMD on domain shift?

Missing references:

[1] Divyansh Kaushik, Amrith Setlur, Eduard H. Hovy, and Zachary Chase Lipton. "Explaining the Efficacy of Counterfactually Augmented Data." In International Conference on Learning Representations. 2021.


--------------------------------------------------------------------------------------
I have read the author response, including the clarifications offered and the additional details promised by the authors that would appear in the final draft. I will keep my score as before and recommend a strong acceptance for this paper and encourage the authors to continue pursuing this line of research.

**Time Spent Reviewing:**

9

---

> ### Author Response · Authors · 2021-08-09
> **Response to reviewer U7gK**
>
> Thank you for the detailed comments and for your support.
>
> “From a practical point of view, requiring one to have knowledge of the underlying causal direction between the covariates and the label of interest can be a limiting factor in practical utility of proposed regularization schemes”
>
> Alas, it’s fundamental to the problem that some causal knowledge is required. That could take the form of access to counterfactual examples, the ability to fully model the structural equation models, or (as in the paper) simply knowing whether Y causes X or vice versa. The last of these seems to be the least onerous requirement. However, it’s certainly true that there are many examples in practice that don’t fit naturally into the considered causal structures, and the method will not be immediately applicable to such instances. This is an important direction for future work, and we hope the techniques in the counterfactual invariance paper provide a template for such work.
>
> “The paper does not offer a discussion of prior work that provides an explanation for the effectiveness of counterfactual invariance as it relates to out-of-domain generalization. I believe this can be easily fixed in a future version. The paper could in general use a better discussion of related work, especially as it relates to Peters et al. (2016).”
>
> As you suggest, we have expanded the related work, particularly including a discussion of the connection to Kaushik et al.
>
> By “Peters et al 2016”, do you mean https://arxiv.org/abs/1501.01332 ? This paper considers the case where we have access to samples drawn from multiple causally compatible domains, and we want to learn a predictor that uses only the causal parents of the label. Per your suggestion, we have expanded our discussion of this line of work in the related work. The connection to counterfactual invariance is a bit nuanced. Retrospectively, it can be understood as: they consider domain generalization across environments defined by arbitrary causal interventions on all variables but Y. They show that a predictor that relies only on the parents of Y is invariant across such environments, and then use this invariance as a learning principle. By contrast, we consider a different set of environments (the “causally compatible” environments of definition 4.1) and show that counterfactually invariant predictors are invariant (in a different sense) across environments. We do not use this invariance as a learning principle in this paper. Additionally, Peters et al does not contain a notion of counterfactual invariance, even implicitly. (The sort of stress testing that inspires the definition seems to have only come into vogue more recently).
>
> “I think the experiments section could use some background on dataset selection, processing, etc.”
>
> The technical aspects are included in the appendix. Following your suggestion, we’ve added some more prose about why we chose these particular data (in short: it gives a natural controlled setting for examining the role of causal vs anti-causal structure) and telegraphed that there are more technical details in the supplementary.
>
> “despite the presence of multiple datasets (representing plausible distribution shifts) for the tasks of how helpful a review is---check Yelp dataset, you can construct various test sets (businesses, restaurants, etc.)---and for natural language inference, the experiments are restricted only to Amazon reviews (from one category) in the former and MNLI in the latter.”
>
> We agree this would be an important and interesting direction. However, the main point of the paper is to formalize and explain stress testing. For our purposes, the role of the domain shift experiments is to illustrate the connection with counterfactual invariance. More elaborate scenarios, where there isn’t a strong a priori reason to expect the domain shift will be causally compatible, would obfuscate the key point about the role of causal vs anti-causal structure.
>
> As discussed in the response to reviewer 92Ro, the complete connection between counterfactual invariance and domain generalization is quite complex, and constitutes enough material for at least a whole additional paper. Extending the counterfactual invariance theory to handle a broader class of domain shifts (and making specific, concrete, and testable predictions about what happens under such shifts) is an important future direction. Such a paper---focused on domain generalization---would certainly benefit from such experiments. However, we think that the current paper does a good job of meeting its goals of elucidating stress testing: explaining the role of causal structure, explaining the implications of stress testing for domain generalization, and testing these implications experimentally.
>
> "Curiosity question: Did you try experiments with regularizing the causally mismatched MMD on domain shift?"
>
> Yes, figure 4 in the paper shows results for both the correct and incorrect regularizers, for both the causal and anti-causal case. As anticipated, regularizing the causally mismatched MMD works worse than regularizing the correct one (in the sense of a worse tradeoff of in-domain accuracy against out-of-domain accuracy) :)

---

> > ### Comment · Reviewer_U7gK · 2021-08-18
> > **Thank you!**
> >
> > Thanks for your response. I'm looking forward to reading the updated version of this paper.

---

### Official Review · Reviewer_ZoEc · 2021-07-16

**Rating:** 6
**Confidence:** 5

**Summary:**

In this work, the authors focus on how to build models that are robust to spurious correlations. The focus of the paper is language datasets. The authors work with two types of DAGs -- a) association of the label with the invariant feature is anti-causal, and b) association of the label with invariant feature is causal. The authors propose a notion of invariance, which they call counterfactual invariance, that they require the predictor to satisfy. Under this notion, the prediction $f$ has to remain invariant under all the potential outcomes that $X$ can have under variations to the spurious feature $Z$. Since the notion of counterfactual invariance is hard to enforce as counterfactuals are not observed, the authors come up with necessary conditions that are then used as proxies to enforce counterfactual invariance. The necessary conditions are different in the two directions causal and anti-causal. Further, the authors analyze the relationship between best counterfactual invariant predictor on train domain and target domain (under appropriate notion of changes allowed in target domain). Finally, the authors establish conditions for min-max optimality of the predictor that is learned.

**Limitations And Societal Impact:**

Yes, the authors have provided the limitations and have stated N/A for negative societal impact.

**Main Review:**

The paper is very nicely written. The application to language models is particularly interesting and novel. I have a few concerns that I highlight below.

1. **Counterfactual invariance notion in fairness**  The authors towards the end in the paper recognize that the notion of counterfactual invariance in an equivalent form already exists [Silva et al.]. Also, from the notion of counterfactual invariance, the necessary conditions that authors identified, which are very well known as demographic parity and equality of odds. Since there is so much overlap with the fairness literature, it is quite important to see the discussion where the notions are introduced and also following the theorem. Also, the authors should explain if they add theory to fairness literature. One would argue that there are papers that establish that equality of odds and demographic parity are not compatible is well-known. I believe that showing how one notion is better suited for one type of DAG and another for another type of DAG might be less well known in fairness literature. Also, another contribution that authors could highlight w.r.t fairness literature might be the connection of counterfactual fairness to demographic parity and equality of odds.

2. **Counterfactual invariance notion in domain generalization** The notion that the authors introduced is very closely related and equivalent under some conditions to the notion of G-invariance in [Robey et al.]. In Robey et al., the authors introduced the notion that the prediction $f(X) = f(G(X,e))$. $G(X,e)$ can be understood as a model that generates the potential outcomes under different environments. I believe a discussion in the same place as to how the notion introduced by the authors and the notion in [Robey et al.] is related is very important.


3. **Reconciling [Robey et al.] and the claims of the authors** In [Robey et al.], the authors learn the model $G$. In the terminology used in the current paper, we can say that the authors learned the model to generate the potential outcomes by observing the variations in the observational data. Finally using the model $G$ the authors enforce $G$-invariance. However, in this paper the authors claim that it is necessarily needed to know the causal or anti-causal direction and then use different penalties accordingly.  How to reconcile that we could learn the model for potential outcomes and not have to worry about two different necessary conditions - - demographic parity and equality of odds. A detailed discussion on whether it is really the case we cannot circumvent the need for knowing the causal direction is very important.


4. **Examples of**  $X_Z^{\perp}$ The authors should actually provide mathematical examples in the paper to show how to decompose $X$ in the two DAGs. Currently, I had to look at the proofs to better understand this and I believe some examples of this can really help the reader.

***

## References

[Silva et al.] Kusner, Matt J., Joshua R. Loftus, Chris Russell, and Ricardo Silva. "Counterfactual fairness." arXiv preprint arXiv:1703.06856 (2017).

[Robey et al.]  Robey, A., Pappas, G. J., & Hassani, H. (2021). Model-Based Domain Generalization. arXiv preprint arXiv:2102.11436.

**Time Spent Reviewing:**

8

---

> ### Author Response · Authors · 2021-08-09
> **Response to reviewer ZoEc**
>
> Thank you for your thorough review and questions.
>
> **Fairness.** As you suggest, we have added a short discussion of the connection to fairness immediately following theorem 3.2. As you say, the contribution here is not that the marginal and conditional penalties are incompatible---this is a simple mathematical fact. Instead, for counterfactual invariance, the point is that different causal structures give rise to incompatible conditions.
>
> With respect to fairness, the contributions here can be understood as that the intuitive stress-test notion of fairness, corresponding to counterfactual invariance, naturally implies the (incompatible) equalized odds or demographic parity conditions, depending on the underlying causal structure. We believe that this is a novel contribution to the fairness literature. This observation also suggests that in many cases one ought to choose the notion of fairness according to the underlying causal structure. Of course, the fairness conditions may have justifications other than through their connection with counterfactual invariance. We leave it to future, fairness focused, work to make the case either for or against the proposition that the relevant ethical criteria are satisfied by choosing the distributional fairness condition implied by counterfactual invariance.
>
> **CF-Invariance in Domain Generalization** For a discussion of the general connection, please see our response to reviewer 92Ro above. In short, we’ve added some clarifying discussion to the paper, and we think that fully articulating the connection between CF-invariance and the domain generalization literature is (easily) a substantive enough subject to be a full paper on its own.
>
> **Robey et al.** This paper assumes that the observed covariates $X$ are generated according to some (learnable) model $g(X_{\mathrm{invar}}, E)$, where $E$ is the environment $X$ is observed in, and $X_{\mathrm{invar}}$ is some unobserved invariant feature set. Then they require that a predictor $f$ satisfies $f(g(X_{\mathrm{invar}}, e)) = f(g(X_{\mathrm{invar}}, e’))$ for all environments $e, e’$.
>
> Now, if we took $X_{\mathrm{invar}} = X_Z^\perp$ and $E=Z$ then this condition would be the same as counterfactual invariance. However, as discussed in our reply to reviewer 92Ro, it is not generally appropriate to take $Z$ and $E$ to be the same---these are conceptually different variables, with distinct causal relationships to the other variables. Accordingly, it’s not particularly natural to think of Robey et al in CF-invariance terms, and it’s not clear what implications the results of the present paper have for their paper. Indeed, they offer a causal model for their setup, figure 2 on page 6 of the arxiv v3, which does not match the causal structures considered in the counterfactual invariance paper. Thus, two papers just aren’t similar enough for the results in one to have simple implications for results in the other.
>
> **Is knowing the causal direction really necessary?** This is addressing the more general question implicit in the Robey et al question: they pose a requirement in terms of counterfactuals, but then don’t go through the process of deriving causal-structure-dependent observable signatures to use for learning. How can that be?
>
> The answer is that you don’t need to consider the causal structure if you actually have access to counterfactual examples, or a sufficiently detailed model for the structural process that you could (learn to) generate counterfactuals. With counterfactuals in hand, you can just enforce directly that the prediction is the same on counterfactual pairs---we discuss papers that take this approach in the related work. Robey et al’s approach can be viewed as learning the structural equation model G and then, via this, accessing the counterfactuals directly. The special structure that makes this possible is detailed structural knowledge of the problem---in their case, encoded in the assumption that image-to-image translation models will capture the structural equation.
>
> (Actually, it’s not totally clear whether Robey et al really works for arbitrary causal structures---that depends on how well the image-to-image translation model generically succeeds. But that’s the assumption that lets their theory be largely agnostic to the gap between counterfactuals and observable signatures depending on causal structure).
>
> **Examples of $X_Z^\perp$**. While $X_Z^\perp$ and $X_Y^\perp$ are not directly instantiated by our proposed approach, one may consider a toy case in which f(X) is a linear bag-of-words model, Z is the review score, and Y is the product category. In this case, terms that are effects of the review score but not the product category, like "good" and "bad", could be viewed as part of $X_Z^\perp$. Terms that are sentiment-neutral but indicative of the product category, like "book" or "movie", could be viewed as part of $X_Y^\perp$. Terms that are causally related to both the sentiment and product category, like "watchable" or "page-turner", would be viewed as part of $X_{YZ}$.

---

> > ### Author Response · Authors · 2021-08-24
> > **Did we fully address your questions?**
> >
> > Do you have any remaining concerns or questions that we can address?

---

> > > ### Comment · Reviewer_ZoEc · 2021-08-25
> > > **Response**
> > >
> > > Thanks for the response. I would appreciate it if you add a discussion regarding comparisons with Robey et al. even if it is in the Supplement due to space constraints. I do not have further questions.

---

### Official Review · Reviewer_92Ro · 2021-07-17

**Rating:** 7
**Confidence:** 4

**Summary:**

This paper presents conditions that are necessary for counterfactual invariant predictions under certain assumptions. They derive these from causal graphs and show the connection to worst-case domain generalization. Experiments on natural language datasets show that the proposed conditions do lead to a decrease in worst-domain error.

**Main Review:**

[Update] Increased my score based on the rebuttal.

The paper presents a nice characterization of causal and anti-causal prediction tasks and the associated independence properties that follow from their respective causal graphs. The examples in Section 2 are useful to understand the main point. Rather than directly regularizing using ground-truth counterfactual examples as in past work, the paper claims to propose "necessary" conditions that can be measured by data. The connection to out-of-domain generalization is good for completeness.

My only reservation is that the paper's key result (the marginal or conditional independence regularizers) is very, very similar to recent line of work in domain generalization once you think of Z as a domain (different domains have perturbations on the non-causal features or their relationship with Y). As a result, it misses an opportunity to address the limitations that have been described for such regularizers.

1) The "marginal regularization"  is the domain invariance condition for representation learning in domain adaptation/generalization (DG) and has been proposed by Muandet (2013), [Li et al](https://openaccess.thecvf.com/content_cvpr_2018/papers/Li_Domain_Generalization_With_CVPR_2018_paper.pdf). Further, recent work has also shown the fundamental issues with this regularization (see [Zhao et al.](https://arxiv.org/abs/1901.09453), [Akuzawa et al.](https://arxiv.org/abs/1904.12543)). The simplest case is where they do not work is when the distribution of Y, P(Y) changes across domains. Perhaps this is why authors needs to assume P(Y)=Q(Y) in Def. 4.1, but this seems a very strong assumption. It will be good to address this literature upfront when the regularizer is proposed---that will also help to justify Def. 4.1 more naturally.

2) The "conditional regularization" term is the class-conditional domain invariance condition (see, e.g.,  [Li et al. (b)](https://arxiv.org/abs/1807.08479), many works actually use the same metric MMD). Can the authors acknowledge this past work upfront in their manuscript since the regularization term is exactly the same? Again, as with 1, recent work has shown the limitations of this regularizer (see [Mahajan et al.](https://arxiv.org/abs/2006.07500)). That work claims that this regularizer is correct only when the distribution of causal features remains the same across domains. It will be good for the authors to discuss their regularizer in light of these results. There is an interesting discussion here, on whether there is a difference in assumptions between the DG literature and this work, or simply a notational difference.

Evaluation: The experiments on natural language tasks are a good addition to the DG literature which mostly focuses on image classification.  For that reason, it may be difficult to compare to the past work in DG. But perhaps the authors can compare their regularizers to simpler DG methods that can work for language tasks too, like contrastive matching over items with the same class but different Z (see [Motiian et al.](https://openaccess.thecvf.com/content_ICCV_2017/papers/Motiian_Unified_Deep_Supervised_ICCV_2017_paper.pdf)) How do the results compare? It is a bit odd to have no baselines on non-causal regularization approaches that have been proposed before.

Overall, I think this paper adds a nice twist to the standard domain generalization/adaptation regularizers and adapts it for robust learning with confounders.



**Time Spent Reviewing:**

2 hours

---

> ### Author Response · Authors · 2021-08-09
> **connections to domain generalization**
>
> Thank you for your thoughtful comments and the additional pointers to the domain generalization literature. As you suggested, we have added a discussion immediately following theorem 3.2 noting that the conditional independence penalties also occur in domain generalization (and fairness) and explaining the connection. We have additionally expanded the related work, including the following discussion clarifying connections with domain generalization.
>
> The connection to domain generalization is rather more subtle than it may first appear. While it is true that interpreting Z as a domain label would make the conditional independence conditions align w/ known DG criterion, this is (usually) not a natural interpretation of Z. In particular, this interpretation of Z actually contradicts the notion of domain generalization considered in the paper. We consider domain transfer across causally compatible domains, where the domains differ due to changes in unobserved confounding or selection variables. In particular, all values of Z are represented in each domain (indeed, in the experimental examples P(Z=0)=P(Z=1)=½ in each domain). Among other things, this means the insights in this paper are not directly comparable to domain adaptation methods that assume access to training samples from multiple domains.
>
> To further clarify the connection, it may help to consider introducing an extra variable E that labels the domain in some fashion that needs to be made precise. For concreteness, consider the anti-causal case where the spurious association between Y and Z is due to an observed confounder U. Now, suppose that E is a cause of U. In this fashion, E labels the distribution of the unobserved confounder, and thus different values of E correspond to different causally compatible domains. Now, if we draw the causal graph corresponding to this situation (apologies, there doesn’t seem to be a way to upload images) then we’ll be able to read off that X_z^\perp \indep E | Y. That is, we arrive at the same conditional independence criterion. However, it’s essentially a coincidence that this matches the criterion we’d use if we’d observed Z---the two variables have totally different causal meanings and interpretations. For example, in the review data case, we might take Z to be review score and E to be the website the review is collected from. The review quality doesn’t need to be counterfactually invariant to the environment, and indeed we wouldn’t expect it to be!
>
> Now, the above does suggest that the counterfactual invariance work can be profitably extended to domain generalization techniques in situations where domain labels are observed. E.g., one could derive conditional independence signatures satisfied by E under the causal/anti-causal and selection/confounding cases---the differences in these would be insightful for when various approaches to domain adaptation succeed or fail. Or, invoking theorem 4.2, we could look for predictors that are invariant across causally compatible domains, noting that CF-invariant predictors satisfy this condition and will also generalize to all other causally compatible domains.
>
> In short: extensions of the counterfactual invariance work seem promising for both methodological improvements in domain generalization, and conceptual and theoretical clarifications. However, such extensions are deep enough that (in our view) they’d clearly constitute an entirely new paper. We think the present paper does a good, self-contained, job of fulfilling the promise of clarifying the importance and means of stress testing including, in particular, the connection of stress testing to domain generalization. The fact that the ideas in the paper have further useful implications for domain generalization appears to be evidence that this is an important and fruitful direction :)

---

> > ### Comment · Reviewer_92Ro · 2021-08-21
> > **Thanks for the clarification**
> >
> > This discussion is helpful and I appreciate the detailed example. Thank you!

---

### Official Review · Reviewer_NhDL · 2021-07-17

**Rating:** 7
**Confidence:** 3

**Summary:**

The paper casts robustness to spurious correlations as a model's dependence on input data that is not stable under stress testing. Stress testing here corresponds to changing part of the input data that a practitioner believes should not change the prediction. The authors term this counterfactual invariance and consider two different causal structures that could've produced the data. Identifying distribution-level implications (as opposed to unit-level) of these causal structures, the authors propose regularization schemes to enforce when learning predictors to ensure that counterfactual-invariance holds (under certain assumptions). The effectiveness of these schemes is shown both theoretically and empirically.

**Limitations And Societal Impact:**

One limitation I can think of for this paper is where the additional variable Z has a large number of categories or is high-dimensional; for example, in the task of paraphrase detection, a possible Z is the number of matching words, length etc. The proposed MMD-based independence method might pose problems because the groups for each possible value-pair of Y, Z may be too small. in the high-dimensional case, most of these groups may have only a single sample.


**Main Review:**

This paper tackles an important problem (of building models robust to spurious correlations) in an interesting and well-motivated way (using independence implications of the assumed causal graph). The main idea of using this independence as regularization in learning representations is well-motivated. The theory supports the claims of generalizing to arbitrary distribution with the same causal structure (and mechanisms) as the one the training data comes from. This, shown in theorem 4.2, is the most illuminating part of the paper, I think.

Empirically, the proposed method seems easy to implement (at least when Z is of small cardinality) and the experiments indicate sizeable improvements. Showing the performance changing for different levels of regularization shed some light onto the workings of the method.

Could the authors explain why in theorem 4.2, the proposed counterfactual invariance property only helps guarantee out of performance generalization for certain kinds of correlations, like the guarantee for the causal-direction graph requires no confounding but works under selection? I suspect this is because the relationship between the label and the covariates can change but a simple of example of how the performance changes would be helpful.

Further, could the authors explain the significance of assuming Z is discrete? It seems like the first lemma depends on it.

Finally, could the authors comment about how one would go about choosing the causal structure over the anti-causal structure (vice versa) for some data at hand?



**Time Spent Reviewing:**

4

---

> ### Author Response · Authors · 2021-08-09
> **Reply to reviewer**
>
> Thank you for your questions and support!
>
> “why in theorem 4.2, the proposed counterfactual invariance property only helps guarantee out of performance generalization for certain kinds of correlations …”
>
> Your intuition about what’s going on is correct. In the causal case, we must rule out the possibility of parents of Y that are not included in the CF-invariant predictor, but which interact with X_z^\perp. If such interacting parents exist, then it is not possible, non-parametrically, to have a predictor that’s invariant across domains because the interactions could be chosen adversarially. We have expanded remark 4.3 to clarify this phenomena.
>
> “Further, could the authors explain the significance of assuming Z is discrete? It seems like the first lemma depends on it.”
>
> Really, Lemma 3.1 requires an assumption that all of the potential outcomes (Y(z))_z are jointly measurable with respect to a single well-behaved sigma-algebra. Assuming that Z is discrete is a conceptually simple sufficient condition, which we chose to avoid burdening the reader w/ measure-theoretic distractions. We have added a remark clarifying this, and noting that the condition we assume is stronger than necessary.
>
> “Finally, could the authors comment about how one would go about choosing the causal structure over the anti-causal structure (vice versa) for some data at hand?”
>
> In short: this judgement relies on human knowledge of the real world. We know that review writers choose a score and then write a review to explain it (the star rating labels the intention of the author)---the problem is anti-causal. We know that helpful votes are based on people reading the review and responding to it---the problem is causal.
>
> Now, it’s not clear in every case what the right answer is. Indeed, in many cases, neither graph is appropriate. Handling such cases requires extension of the ideas described in the paper. This is an important direction for future work. However, the cases considered in the paper do cover many practical applications, and, together, illustrate the fundamental role of the underlying causal structure.

---

### Decision · Program_Chairs · 2021-09-27

**Decision:**

Accept (Spotlight)

**Comment:**

All reviewers have given very positive reviews with positive revisions after the discussion.

Briefly, the notion of counterfactual invariance discussed, its relation to stability against spurious variation and different conditions/regularization depending on the DAG model that is needed to enforce it have been well appreciated. Empirical results on text have been satisfactory as well.

There remains some minor concerns which authors have promised to address in their response. I encourage the authors to add additional material that they have promised to various reviewer concerns.